# Study of the Phytoextraction and Phytodegradation of Sulfamethoxazole and Trimethoprim from Water by *Limnobium laevigatum*

**DOI:** 10.3390/ijerph192416994

**Published:** 2022-12-17

**Authors:** Klaudia Stando, Aleksandra Czyż, Magdalena Gajda, Ewa Felis, Sylwia Bajkacz

**Affiliations:** 1Department of Inorganic, Analytical Chemistry and Electrochemistry, Faculty of Chemistry, Silesian University of Technology, B. Krzywoustego 6 Str., 44-100 Gliwice, Poland; 2Biotechnology Centre, Silesian University of Technology, B. Krzywoustego 8 Str., 44-100 Gliwice, Poland; 3Environmental Biotechnology Department, Faculty of Power and Environmental Engineering, Silesian University of Technology, Akademicka 2 Str., 44-100 Gliwice, Poland

**Keywords:** phytoremediation, sulfamethoxazole, trimethoprim, phytoextraction, phytodegradation, transformation products, LC-MS/MS

## Abstract

Phytoremediation is an environmentally friendly and economical method for removing organic contaminants from water. The purpose of the present study was to use *Limnobium laevigatum* for the phytoremediation of water from sulfamethoxazole (SMX) and trimethoprim (TRI) residues. The experiment was conducted for 14 days, in which the loss of the pharmaceuticals in water and their concentration in plant tissues was monitored. Determination of SMX and TRI was conducted using liquid chromatography coupled with tandem mass spectrometry. The results revealed that various factors affected the removal of the contaminants from water, and their bioaccumulation coefficients were obtained. Additionally, the transformation products of SMX and TRI were identified. The observed decrease in SMX and TRI content after 14 days was 96.0% and 75.4% in water, respectively. SMX removal mainly involved photolysis and hydrolysis processes, whereas TRI was mostly absorbed by the plant. Bioaccumulation coefficients of the freeze-dried plant were in the range of 0.043–0.147 for SMX and 2.369–2.588 for TRI. Nine and six transformation products related to SMX and TRI, respectively, were identified in water and plant tissues. The detected transformation products stemmed from metabolic transformations and photolysis of the parent compounds.

## 1. Introduction

Sulfonamides (SAs) are abundant in the aquatic environment [1,2]. However, SAs are not completely biodegradable in biological [3,4] and chemical [5,6] wastewater treatment processes and are reintroduced into the aquatic environment. Moreover, SAs are transformed by conventional treatment processes into transformation products (TPs) with unknown physicochemical and potentially toxic properties [5,7,8]. Out of two SAs approved as human medicine, sulfamethoxazole (SMX) in combination with trimethoprim (TRI) is one (SMX + TRI) [9]. SMX + TRI, which appears under the trade names Bactrim^®^, Biseptol^®^, and Trimesolphar^®^, is used to treat bacterial infections of the urinary tract, respiratory tract, and gastrointestinal tract. According to previous studies, SMX and TRI have been detected post-wastewater treatment with activated sludge, and in river water after wastewater discharge [10]. The concentration of SMX and TRI in the river water after wastewater discharge was 16.9 and 150.0 ng L^−1^, respectively [10]. In addition, three SAs were identified in aquatic environmental monitoring studies of five water bodies in Silesia (Poland): SMX (0.1–75.8 ng L^−1^), SFD (<0.1 ng L^−1^), and SFP (0.1–38.8 ng L^−1^) [2]. The development of a low-cost, effective, and environmentally friendly method of removing SA micropollutants from the water after the wastewater treatment process is necessary to prevent recontamination of the environment.

Phytoremediation is commonly used in the treatment of soil and water to remove heavy metal residues [11,12]. Plants capable of absorbing large amounts of selected heavy metals are referred to as hyperaccumulators, and their mechanism of action involves the formation of stable phytochelatins with metal ions, allowing their internal accumulation through the shoots and roots of the plant [11,12]. The phytoremediation process is being increasingly used to remove pharmaceuticals from the soil and water [13,14]. However, the application of phytoremediation requires a proper understanding of the various factors involved and its mechanism. Phytoremediation consists of several parallel processes: phytoextraction, phytoaccumulation, phytodegradation, and phytoevaporation [15].

The efficiency of the phytoremediation process for selective contaminant removal is influenced by physiological, biological, and physicochemical factors, such as light intensity, temperature, pH, plant species, the presence of microorganisms in the aquatic/soil environment, duration of exposure, and stomatal conductance [16]. The most important factors that influence the effectiveness of phytoremediation are the metabolism specific to the plant species, and the type and concentration of contaminants. Therefore, selecting the appropriate plant is necessary to achieve contaminant remediation [16,17]. In the case of phytoremediation, the plant selected should be easily available, exhibit a potential for bioaccumulation of contaminants, be highly resistant to environmental factors, and possess high biomass potential [17]. Additionally, decorative plants should be used for the phytoremediation process, as they are not reintroduced into the human digestive system and can serve an aesthetic function [15]. Water phytoremediation studies are commonly conducted using free-floating plants, such as *Pistia stratiotes*, *Salvinia molesta*, and *Eichhornia crassipes*, due to their high availability, high biomass potential, and easy harvesting after the remediation process [17]. Despite the known potential of phytoremediation towards pollutants by *Salvinia molesta* and *Eichhornia*, their disadvantage includes high invasive potential [18,19]. The European Union has listed them as invasive alien species whicha pose a threat [20], hence it is necessary to find other plant species that are potentially less harmful to the environment.

The purpose of this study was to evaluate the accumulation of SMX and TRI by the plant species *Limnobium laevigatum* (*L. laevigatum*) when introduced to contaminated water under controlled conditions. *L. laevigatum* is a free-floating, perennial, aquatic macrophyte with short petioles and round leaves. This species was selected due to its rapid population growth rate and high efficiency (nearly 80%) in reducing chemical oxygen demand (COD) [21]. Reports have shown that *L. laevigatum* is a hyperaccumulator of heavy metals (Pb, Cr, Ni, Zn), and the presence of contaminants do not have a detrimental effect on its survival [21,22]. Information related to the utilization of *L. laevigatum* for phytoremediation of organic pollutants is limited. To date, the effectiveness of phytoremediation of organic contaminants from water using this plant has been tested for bisphenol A, 17-α-ethinylestradiol, and estrone [23]. Our study employed liquid chromatography coupled with a tandem mass spectrometer (LC-MS/MS) to determine SMX and TRI residues in water and plant tissues. In addition, untargeted analysis was performed to identify TPs of SMX and TRI and determine their transformation pathways in plant tissues and water samples. Determination of transformation paths in laboratory conditions allows us to determine which TPs are formed in the aquatic environment and which are the result of plant metabolism. The experiment was conducted over a 14-day period under controlled conditions (constant temperature, irradiation). The obtained results expand current knowledge of the application of *L. laevigatum* for the removal of pharmaceutical micropollutants from the aquatic environment, which can accumulate over a long time. As part of the work, the main factors affecting the removal of SMX and TRI from water in the phytoremediation process were determined.

## 2. Materials and Methods

### 2.1. Standards, Chemicals, and Materials

Analytical standards of sulfamethoxazole (SMX) and trimethoprim (TRI) were purchased from Sigma-Aldrich (St. Louis, MO, USA). Hypergrade acetonitrile, formic acid, and water were purchased from Merck (Darmstadt, Germany). Analytical-grade methanol, hydrochloric acid, sulfuric acid, and disodium phosphate were purchased from CHEMPUR (Piekary Śląskie, Poland). Analytical-grade acetic acid was purchased from POCH S.A. (Gliwice, Poland). Analytical-grade acetonitrile was purchased from Eurochem (Tarnów, Poland). Analytical-grade citric acid, formic acid, and acetic acid were purchased from ChemLand (Stargard, Poland). Analytical-grade sodium hydroxide was purchased from POCH S.A. (Gliwice, Poland). Oasis HLB cartridges of varying masses of sorbent (500 mg, 6 mL; 400 mg, 6 mL; 200 mg, 3 mL) from Waters (Eschborn, Germany) were used for solid-phase extraction.

### 2.2. Characteristics and Preparation of Limnobium Laevigatum Cultivation in Hydroponic Conditions

*L. laevigatum* seedlings were obtained from a local aquarium shop in Gliwice, Poland. The plants were transferred to the laboratory, where the roots were rinsed with distilled water and transferred to a 10 L aquarium from Diversa (Prudnik, Poland). Each aquarium contained 8 L of tap water, and the plants were allowed to acclimatize for 7 days before starting the experiment. All plants were cultivated under dual-head LED grow light ZEED (Świętochłowice, Poland), which simulated natural sunlight. The lamp covered the full wavelength range of light, i.e., 400–840 nm, which was the most efficient for enhancing the photosynthesis performance. The lamp operation mode was 12 h (12 h-light, 12 h-dark).

After the plant acclimatization period, the phytoextraction experiment was performed. Under laboratory conditions, 5 experimental aquariums were prepared in three variants: (1) tap water and plants supplemented with SMX or TRI (PWAMs; 3 experiments); (2) tap water without plants supplemented with SMX or TRI (WAMs: 1 experiment); and (3) tap water and plants without antimicrobial supplementation (PW; 1 experiment). The initial antimicrobial (SMX or TRI) concentration introduced to the water was 1 µg L^−1^. Phytoremediation experiments were carried out separately for each antimicrobial. PWAMs and PW contained 8 mature *L. laevigatum* plants. The experiment was carried out over a 14-day period. *L. laevigatum* seedlings were not grown in in vitro conditions, therefore, it was necessary to perform a blank test (PW) to exclude contamination of the seedlings by SMX or TRI.

### 2.3. Sampling and Preparation of the Samples

The Baker^®^ SPE-12 G system (vacuum manifold for 12 cartridges, Witko, Łódź, Poland) was applied in the study. The SPE system was connected to a vacuum pump (LABOPORTTM PowerDryTM Vacuum Pump, KNF Neuberger, Trenton, NJ, USA) to obtain the required flow rate. The experiment measured the loss of SMX and TRI in water and their bioaccumulation capacity in the plant. Water samples were taken on the day of starting the experiment (day 0) and daily for the first 7 days, and then on days 9, 11, and 14 of the experiment. Plants were harvested on day 0 and then on days 7 and 14 of the experiment. SMX and TRI from the water samples were extracted using solid-phase extraction (SPE), and solid-liquid extraction (SLE) was used for the plant samples. The extraction conditions were described in our previous work and described below [2,10,24].

#### 2.3.1. SMX and TRI Extraction from Water Samples

The Waters OASIS^®^ HLB cartridge (500 mg, 6 mL) was preconditioned each time with 6 mL of methanol, 0.1 M HCl, and distilled water adjusted to pH = 4 with H_2_SO_4_. Then, 100 mL of water was filtered through a glass fiber filter (MN GF-1, Machery-Nagel, Düren, Germany) and adjusted to pH = 3.90–4.10 with H_2_SO_4_. The sample was passed through the cartridge with a flow rate of 3 mL min^−1^ using a vacuum pump. The cartridge was allowed to dry for 30 min and the analytes were eluted. Elution was carried out using 6 mL of methanol and the extract was evaporated to dryness. Finally, the residue was re-dissolved in 1 mL of methanol and 3 μL of the extract was injected into the chromatographic system (described in Section 2.4). The analysis was repeated nine times.

#### 2.3.2. SMX and TRI Extraction from Plant Samples

After collection, the aquatic plants were dried with a paper towel to remove the residual water, placed in polypropylene containers, and then freeze-dried. Frieze-drying was carried out under 0.035 mbar of pressure at −50 °C ALPHA 1–2 Ldplus (CHRIST, Osterode am Harz, Germany). Freeze-dried plant material was homogenized using an electric grinder MK70 DOTT (ELDOM, Katowice, Poland). An amount of 0.5 g of ground freeze-dried sample was extracted twice with 10 mL of methanol. A single SLE was performed for 30 min at 750 rpm using Vibramax 100 (Heidolph, Schwabach, Germany). For each repetition, the samples were centrifuged for 10 min at 8000 rpm using Hermle LaborTechnik Z 323 Centrifuge (Hermle, Wehingen, Germany), and the supernatants were combined. The sample was evaporated to dryness under a stream of nitrogen and dissolved in 1 mL of 0.1% formic acid in H_2_O:MeOH (1:1; *v*/*v*) before analysis.

### 2.4. Instrumentation and Analytical Conditions

Dionex UltiMate 3000 HPLC system (Dionex Corporation, Sunnyvale, CA, USA), which consisted of an UltiMate 3000 rapid separation pump, autosampler, and thermostatted column compartment, was used for the analysis. To control the chromatography system, Dionex Chromeleon TM 6.8 software was used. The chromatographic analysis of SMX and TRI was carried out using a Kinetex Core-Shell C18 column (75 mm × 2.1 mm, 2.6 μm, Phenomenex, Torrance, CA USA) and isocratic elution. The mobile phase consisted of a mixture of acetonitrile (solvent A) and 0.1% formic acid in water (solvent B), and the composition of the mixture was 80:20 (A:B; *v*/*v*). The column was maintained at 30 °C and the flow rate was 0.6 mL min^−1^. The injection volume was 3 μL.

The HPLC system was coupled with an AB Sciex Q-Trap^®^ 4000 mass spectrometer (AppliedBiosystems/MDS SCIEX, Foster City, CA, USA). Analyst 1.4 software was used to control the mass spectrometer work. SMX and TRI were analyzed in the positive ionization mode. Targeted analysis was performed in the multiple reaction monitoring mode (MRM). MRM transitions and optimized MS parameters (i.e., the declustering potential (DP), collision energy (CE), collision cell exit potential (CXP), and entrance potential (EP)) for SMX and TRI were described in our previous work [10]. The ion source parameters were optimized by flow injection analysis as follows: ion spray voltage (IS) = 4000 V, temperature (TEM) = 500 °C, collision gas (CAD) = medium, curtain gas (CUR) = 10 psi, ion source gas 1 (GS1) = 60 psi and ion source gas 2 (GS2) = 50 psi. Representative chromatograms obtained in MRM acquisition mode for SMX and TRI are shown in Appendix A.

The same column, temperature, mobile phase, and sample injection volume were used for the non-target analysis, as described in Section 2.4. A gradient elution program was used for the separation of the TPs. The chromatographic separation started with 10% solvent A, and increased linearly for 5 min to 35%. From 5 to 7 min, solvent A content increased to 85%. From 7.1 min, the equilibration of the chromatographic system to the initial conditions began. The total analysis time was 10 min. The samples were analyzed using LC-MS/MS by three methods: pseudo-MRM mode (p-MRM) with positive ionization, enhanced mass spectrometry (EMS), and enhanced product ion (EPI). The ion source parameters were the same as those used in the case of the developed LC-MS/MS method.

### 2.5. Determination of SMX and TRI and Their TPs Using LC-MS/MS

SMX and TRI phytoremediation experiments with water containing *L. laevigatum* were carried out in separate aquariums for each drug. Both experiments began at the same time under the same laboratory conditions. SMX and TRI were extracted from water samples on the day of collection according to the procedure described in Section 2.3. After freeze-drying, the plant samples were stored in a freezer at −15 °C for no longer than a week, and then the extraction procedure was applied. The samples were analyzed using LC-MS/MS method in MRM mode as described in Section 2.4. The content of SMX and TRI in the samples was determined using the calibration curves. The ability of the plant to accumulate pharmaceuticals was assessed using the bioaccumulation factor (BAF). BAF is described in Equation (1) as the ratio of the pharmaceutical concentration in the plant and water.
(1)$${\mathrm{BAF}}_{i}\;=\;\frac{C_{i}}{W_{i}}$$
where *C_i_* is the concentration of pharmaceutical in the plant [mg kg^−1^], and *W_i_* is the concentration of the same pharmaceutical in water [mg kg^−1^] at 25 °C (ρ_water_=997.07 kg m^−3^).

Screening analysis of TPs was performed by comparing the results of p-MRM analyses with literature data. Based on the literature, a list of SMX and TRI TPs that were most often found in environmental samples was prepared [10]. The intelligent data acquisition (IDA) mode was used for non-targeted analysis. The results obtained in EMS–IDA–EPI mode confirmed the presence of TPs detected in p-MRM mode. Masses ranging from 100 to 500 Da were adopted as IDA criterion. A retrospective spectral analysis was also performed to detect TPs not described in the literature.

### 2.6. Method Validation

The development and validation of the method for water and plant samples were previously reported [2,10,24]. The validated LC-MS/MS method was successfully used to determine SMX and TRI in water and plant samples.

## 3. Results and Discussion

### 3.1. Phytoremediation Potential of L. laevigatum

#### 3.1.1. Determination of SMX and TRI in Water

The degree of pharmaceutical contaminant removal from water during the phytoremediation process under controlled conditions was due to three phenomena: photolysis, hydrolysis, and plant sorption. SMX and TRI are not volatile [25], so their removal was not related to evaporation from the water during the experiment. Table 1 shows the loss of SMX from water over 14 days, taken from PWAMs (hydrolysis + photolysis + plant sorption) and WAMs (hydrolysis + photolysis). No residues of SMX and TRI were detected in the water from the PW.

The results showed that 96.0% of the initial SMX content (1.0 μg L^−1^) was removed after 14 days, in which 52.9% was removed after 2 days. Photolysis and hydrolysis had the greatest effect on SMX degradation, causing 83.8% loss of SMX after 14 days. SMX sorption by the plant was in the range of 4.5–31.8%, and the highest removal efficiency by *L. laevigatum* was observed between days 3 and 5 (28.8–31.8%). The pH of the solution and water matrix composition played key roles in the photolytic and photocatalytic degradation of SMX in water. According to the literature, pH determines the ionic form in which the compound exists in solution and therefore affects changes in the surface charge density of the molecule and the extent to which it absorbs light [26,27]. pKa1 and pKa2 values of SMX are 1.85 ± 0.30 and 5.60 ± 0.04, respectively, with SMX above pH > 5.6 being in the anionic form, while for pH ranging from 1.8 to 5.6, the neutral form predominates [27]. The obtained results of an OECD 111 test indicated that the hydrolysis rate of SMX was the highest at pH = 4 and lowest at pH = 9, which stemmed from the lower sensitivity of the anionic forms to the hydrolysis process compared to the neutral and cationic forms [28]. Subsequent studies confirmed this relationship; under acidic to weakly basic conditions (pH = 4 to 8), the photodegradation rate constant (k) of SMX decreases with increasing pH from 0.621 min^−1^ (pH = 4) to 0.394 min^−1^ [26]. During the experiment we conducted, the pH of the water varied in the range of 6.0–7.0, indicating the predominance of the anionic form of SMX throughout the experiment. According to the literature, SMX in its anionic form is repelled from plant roots in the soil, which may affect the root uptake of SMX by *L. laevigatum* in water [29].

Table 2 shows the percentage loss of TRI in water in the phytoremediation experiment with *L. laevigatum* (PWAMs) and light (WAMs). After 14 days of phytoremediation, 75.4% of the initial TRI content was removed from the water (1.0 μg L^−1^). The main mechanism of TRI removal involved sorption by the plant, whose efficiency was highest between days 2 and 7 (51.4–58.6%). TRI degradation by photolysis and hydrolysis was linear, and the daily loss of TRI in water taken from WAMs ranged from 0.6% to 12.2%. This result was consistent with the characteristics of the TRI compound, which is a stable drug that undergoes slow degradation in environments with extreme pH values, while at elevated temperatures or under sunlight, stable TPs are formed [30]. Sirtori et al. confirmed that under photolysis conditions in distilled water, 50% of TRI was lost after 780 min of exposure, while in salt water the time increased to 1400 min [31].

A comparison study of the obtained results revealed that TRI was better assimilated by the plant than SMX. In the case of SMX, photolysis and hydrolysis were the dominant factors responsible for its degradation [32,33]. This was consistent with the literature, as the half-lives of SMX and TRI in water samples from wastewater treatment wetlands were 1 h and 2.3 h, respectively, and 1.7 h and 24 h for the non-wetland matrix, respectively [33]. The bioavailability of pharmaceutical contaminants depended on the plant species, its specific metabolism, and culture conditions (soil/hydroponic). Reports have described that, under hydroponic conditions using Brassica rapa var. pekinensis and Brassica rapa, higher uptake of SMX than TRI from water was observed [34]. However, *L. laevigatum* showed the opposite trend, where TRI was favorably taken up by plants.

#### 3.1.2. Bioaccumulation Study of SMX and TRI in *L. laevigatum* Tissues

Table 3 summarizes the determined contents of SMX and TRI in the plant samples taken from three parallel cultures of PWAMs and the bioaccumulation rates of contaminants after 7 and 14 days of phytoremediation. SMX and TRI were not detected in the plant samples taken from PW. It was observed that both pharmaceuticals were absorbed by the plant, with SMX concentrations in tissues significantly lower than TRI under the same conditions. The condition of the plants throughout the experiments in the presence of the pharmaceuticals was satisfactory, and no plant death or root dwarfing was observed. Furthermore, SMX concentration in plant tissues on day 7 of the experiment was higher (146.6 ng g^−1^) than on day 14 (42.6 ng g^−1^), which was possibly due to the metabolism of SMX in the plant, and promoted TPs formation. The rapid metabolization of SMX was observed for Arabidopsis thaliana where, after 10 days of exposure, only 1.1% of SMX was present in the form of the parent compound, and the remainder as N-glycolyzed TPs [35]. BAF_FW_ and BAF_DW_ values for SMX were lower than those for *B. rapa chinensis* (0.1–0.4; BAF_FW_), *I. aquatica* (<0.1; BAF_FW_) [32] and comparable to that for *Brassica oleracea* (0.085–10.92; BAF_DW_) [34].

TRI concentration in plant tissues taken from PWAMs was lower on day 14 (72.0 ng g^−1^) by 8.2% compared to day 7 (78.4 ng g^−1^). The easy bioavailability of TRI by *L. laevigatum* is promising, as it is toxic to most plant species [36,37]. In the reported studies conducted on sweet oat, rice, and cucumber seeds, TRI at EC50 values of 86 mg L^−1^, 118 mg L^−1^ and >300 mg L^−1^, respectively, inhibited plant germination [36]. In an acute toxicity study of TRI from the willow tree, the toxic effect of TRI (pKa = 7.2) was dependent on its ionic form, with the inert form being more toxic to the plant than the cationic form [37]. The results of the toxicity test conducted using *Chlorella vulgaris* indicated that TRI was less toxic (EC50 = 90.86–384.71 mg L^−1^) than SMX (EC50 = 0.95–9.31 mg L^−1^) [38], which could explain the faster metabolism of SMX than TRI by *L. laevigatum* between days 7 and 14. The bioaccumulation coefficient of TRI did not change significantly after 14 days. The determined BAF_DW_ of TRI in the study (0.07; 0.078) was higher than BAF_DW_ obtained for cabbage leaves (0.0383) [34].

Based on the obtained data, *L. laevigatum* did not tend to hyperaccumulate any of the tested compounds. The BAF of both SMX and TRI were similar or lower than the literature values [32,34]. The differences in SMX and TRI concentrations (Table 3) were consistent with the observed trend of drug removal from water (Table 1 and Table 2), in which TRI sorption was dominant, whereas for SMX, photolysis and hydrolysis were dominant.

### 3.2. Identification of SMX and TRI TPs in Water and Plant Tissues

Once absorbed by the plant, pharmaceuticals are metabolized with the formation of TPs [35]. Appendix A shows the structures of the identified SMX and TRI transformation products using p-MRM–IDA–EPI and EMS–IDA–EPI methods. TPs of SMX and TRI were determined not only in the plant but also in water, where they were released by the roots or formed by degradation under the influence of abiotic factors (photolysis and hydrolysis) [39]. Table 4 shows the formation of TRI TPs over 14 days, and Figure 1 shows their transformation pathways.

Six TRI TPs were identified: three TPs formed by hydroxylation of the pyrimidine ring (TRI325) or 3,4,5-trimethoxyphenyl ring (TRI323, TRI307), and two TPs formed by mono- and di-oxidation of TRI (TRI291, TRI305). TRI323 exists in the form of three isomers, since hydroxylation occurs at C-2 and C-6 position of the methylene ring and methylene bridge [40]. The structures of the isomers are shown in Appendix A. The mono-hydroxylated trimethoprim derivative (TRI307) exists in the form of three structural isomers, where the hydroxyl group is attached at C-2 position of the nonsynonymous ring, methylene bridge, or the amine group in the pyrimidine ring [41]. TRI291 exists in five isomeric forms, with (2,4-diaminopyrimidine-5-yl)-(4-hydroxy-3,5-dimethoxyphenyl)methanone being the most likely structure [41] (Figure 2a). In the structure of TRI325, amide-imide tautomerization occurs at the hydroxylated pyrimidine ring, as shown in Appendix A. TRI171 is formed by C-C bond cleavage between the pyrimidine ring and 1-methyl-3,4,5-trimethoxybenzene and oxidation of the amino groups (Figure 2b).

In the case of plant tissues taken from PWAMs, three of six TPs were identified, in which TRI323 and TRI307 were observed after only 7 days, and TRI305 after 14 days. It is known that after uptake, pharmaceuticals can be metabolized by the plant with the formation of phase I, II and III metabolites, similar to human metabolism [42]. In our experiment, the three identified TPs were formed by hydroxylation and oxidation reactions, which displayed the characteristic of phase I of plant metabolism. TRI-hydroxylated and -oxidized TPs were commonly detected in reaction mixtures of photodegradation [39,43], electrochemical [41], thermo-active oxidation [40], and biological wastewater treatment [44] processes. TRI305 and TRI307 were identified as phase I products of plant metabolism in lettuce (*Lactuca sativa* L.) [45], whereas TRI305 alone was identified in four species of microalgae [46]. TRI171 was only detected in water from PWAMs, and, to date, TRI171 has been observed in photocatalytic oxidation in the presence of Fe(VI) [47]. Its presence in water was probably related to the photolytic degradation of TRI. The existence of TRI171 only in the water from PWAMs was justified by the release of substances into the water by the plants. Compounds and micronutrients released by the plant may affect TRI photolysis, hence it was not observed in water from WAMs. TRI325 and TRI291 were detected only in water from WAMs after 7 and 9 days of the experiment, respectively. The photolytic degradation of TRI with the formation of TRI325 and TRI291 was previously confirmed in the literature, where they were detected by electrochemical oxidation [41], the Fenton process [48], and photocatalysis [49].

Nine SMX TPs were identified in water, formed by hydroxylation (SMX288, SMX304), S-N bond cleavage (SMX173, SMX179), N-O bond cleavage (SMX256), isoxazole ring rearrangement (SMX254), isoxazole ring cleavage (SMX216, SMX246), and SMX dimerization (SMX445). Table 5 shows the formation of SMX TPs over 14 days, and Figure 3 shows their transformation pathways.

Four TPs (SMX304, SMX254, SMX445, SMX246) were formed by metabolic changes in the plant from PWAMs. All four TPs were detected after 7 days in the plant, while only SMX304 and SMX445 were present after 14 days. SMX304 was formed via the attachment of three hydroxyl groups to SMX. Hydroxylation is a reaction typical of phase I of plant metabolism; monohydroxylated TPs of SMX have been identified in *Arabidopsis thaliana* [35] and *Chlorella pyrenoidosa* microalgae [50]. SMX254 is the isomerization product of the isoxazole ring of SMX, which has been identified in advanced oxidation processes [51]. SMX445 was formed by dimerization of two SMX molecules. SMX246 was formed via the opening of the isoxazole ring, followed by oxidation. Both SMX445 and SMX246 were identified by electrochemical degradation of SMX [52], although metabolic changes in the plant should be considered. Another two TPs (SMX216, SMX179) were detected in both plant tissues and water taken from PWAMs. SMX179 was formed via S-N bond cleavage and attachment of a hydroxyl group to form (5-methyl-1,2-oxazol-3-yl)amidosulfonic acid (Figure 4a). SMX179 was first detected in the reaction mixture after oxidation with sodium hypochlorite [51], however, it could form under environmental conditions. SMX179 was present both in water from PWAMs and WAMs, suggesting that this TP was generated by photolysis of SMX, and was subsequently absorbed by the plant. SMX216 was formed by opening the isoxazole ring, followed by transformation into a carbaimide group. Reports have shown that SMX216 can be produced by microbial activity [53,54]. In our previously reported study, SMX216 was detected in parsley roots exposed to SMX for a period of 3 months [24]. Therefore, it remains unclear if SMX216 was absorbed from water or formed directly in the plant and released through the root pathway.

Three TPs (SMX173, SMX256, SMX288) were identified in water samples collected from PWAMs and WAMs, but were not present in plant tissues. SMX288 was identified only in water taken from PWAMs after 7 days, and was produced by double hydroxylation of the isoxazole ring (Figure 4b). SMX288 is formed by both abiotic agents (ozonation [6,51], radiation [55]) and microorganism activities [53]. SMX288 was observed only in water collected from PWAMs, hence, their secretions affected SMX photolysis. SMX173 is produced through S-N bond cleavage in SMX, followed by 4-aminobenzosulfonate formation. SMX173 can form under the influence of both biotic and abiotic factors, where its presence was observed after microbial degradation processes of SMX [53,54] and the photo-Fenton process [56]. SMX173 was identified in water collected from both PWAMS and WAMs after 4 days. However, it was not detected in plants, suggesting that it was generated via photolysis of SMX in water. SMX256 is produced through N-O bond cleavage in the isoxazole ring and was identified only in water taken from WAMs. A SMX photolysis product was present in water between day 6 and day 14. This product was also detected in the electrochemical degradation of SMX [52].

Based on the retrospective analysis of mass spectra, six TPs of TRI and nine TPs of SMX were identified in plant tissues and water. Transformation of SMX or TRI in phytoremediation was due to plant metabolism or photolysis. Four SMX derivatives (SMX304, SMX254, SMX445, SMX246) and three TRIs (TRI323, TRI305, TRI307) were identified as products of plant metabolism, which were present only in plant tissues. The remaining TPs were formed by photolysis (TRI171, TRI291, TRI325, SMX256, SMX288, SMX179, SMX173) or secreted by the plant through the root route (SMX216). It is well known that the composition of the water matrix and pH [57,58,59] affect the photodegradation process of contaminants. Presumably, the matrix components released by the plant through the root pathway influenced the photolysis of SMX and TRI. SMX288 and TRI171 were detected only in water in which the plants were grown, but were not present in the plant tissues or water from the control sample.

## 4. Conclusions

SMX, in combination with TRI, is one of the most widely used drugs due to its broad spectrum of action against bacteria and some protozoa. Conventional water treatment methods are not effective in complete SMX and TRI micropollutant removal and these are reintroduced into the environment. Phytoremediation is a promising and environmentally friendly alternative to traditional biological and chemical treatment methods. The ability of plants to bioaccumulate pollutants and transform them through metabolism is an interesting research area with great application potential.

The presented study involved the examination of the phytoremediation of SMX and TRI from water under laboratory conditions over a 14-day period. *L. laevigatum* was used for phytoremediation due to its high biomass potential, rapid growth and development, and easy harvesting of the plant. The phytoremediation process removed 96.0% of SMX and 75.4% of TRI from water. The main mechanism for SMX removal was photodegradation, which removed 83.8% of SMX.

In contrast to SMX, the loss of TRI was mainly due to plant sorption, which resulted in the loss of 58.6% of TRI from water after 3 days. SMX concentration in the leaves was higher on day 7 compared to day 14, which was due to the plant metabolizing the drug during the experiment. This theory was supported by the identification of six SMX TPs in the plant tissues, including four TPs (SMX304, SMX254, SMX445, SMX246) formed in reactions of Phase I of their metabolism. TRI concentration in plant tissues also decreased during the experiment, although to a lesser extent than for SMX. Two TPs of TRI were detected in tissues after 7 days (TRI323, TRI307), while TRI305 was detected after day 14. Based on the calculated BAF, SMX and TRI were similar or lower than the literature data for other plant species. Hence, *L. laevigatum* did not show a tendency to hyperaccumulate SMX and TRI. In the screening analysis, a total of six TPs of TRI and nine TPs of SMX were identified and their origin was determined. Transformation of the parent compounds occurred through photodegradation in water and plant metabolism.

This is one of the first publications to comprehensively describe the process of water purification from pharmaceutical micropollutants using plants. The presented research includes analysis of the process of SMX and TRI removal from water, accumulation in the plant, and transformation of these compounds during the experiment. The main factors affecting the removal of selected drugs in the phytoremediation process were determined. This work fills the research gap regarding SMX and TRI transformation products formed in the aquatic environment. Importantly, the determination of the tissue distribution of TPs made it possible to determine which of them were formed as a result of plant metabolism.

## Figures and Tables

**Figure 1 ijerph-19-16994-f001:**
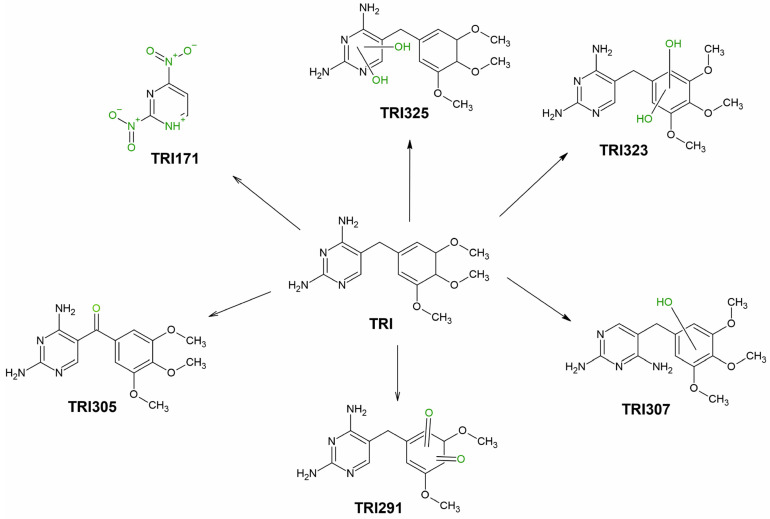
TRI transformation path in the phytoremediation process (The change in the TRI structure is marked in green color).

**Figure 2 ijerph-19-16994-f002:**
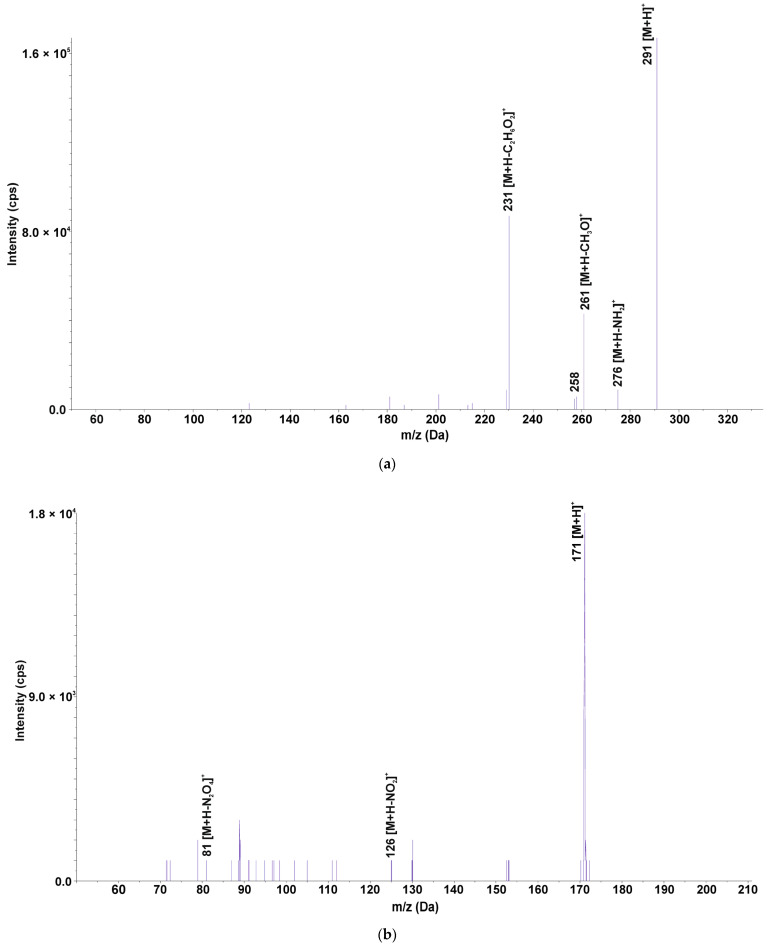
Mass spectra registered in EMS–IDA–EPI mode of the selected TRI TPs: (**a**) TRI291 (*m*/*z* 291) (**b**) TRI171 (*m*/*z* 171).

**Figure 3 ijerph-19-16994-f003:**
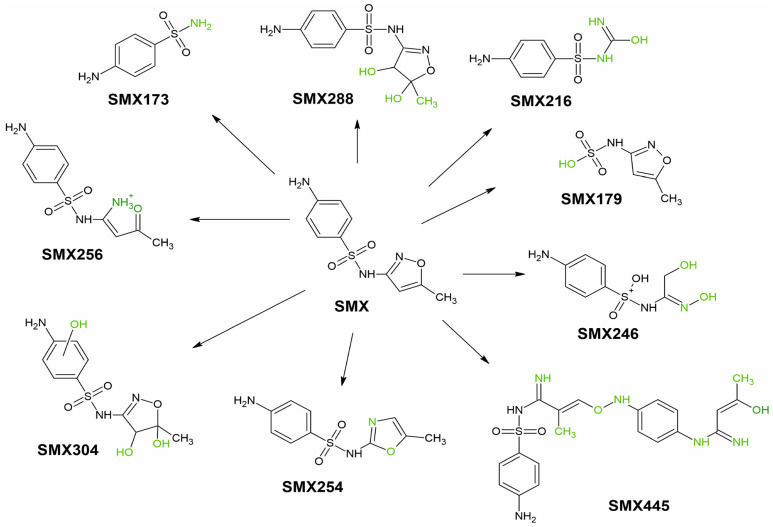
SMX transformation path in phytoremediation process (The change in the SMX structure is marked in green color).

**Figure 4 ijerph-19-16994-f004:**
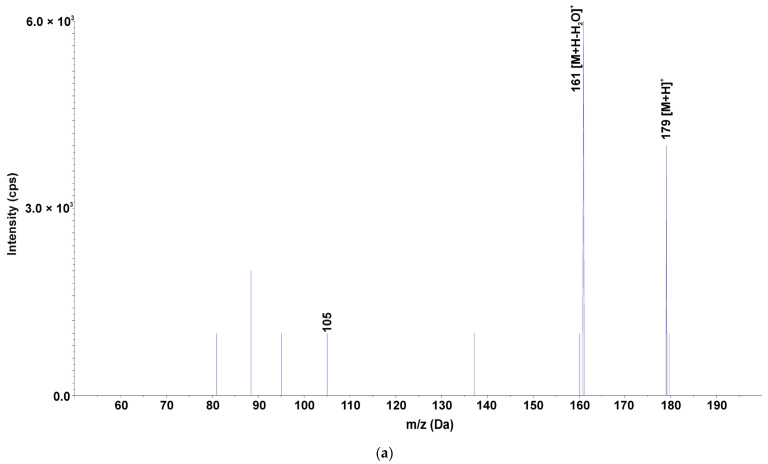
Mass spectra registered in EMS–IDA–EPI mode of the selected SMX TPs: (**a**) SMX179 (*m*/*z* 179) (**b**) SMX288 (*m*/*z* 288).

**Table 1 ijerph-19-16994-t001:** Removal (%) of SMX from water through photolysis, hydrolysis and sorption by the plant.

Day	Factors		
	PWAM_Av._ Photolysis + Hydrolysis + Plant Sorption (RSD) [%]	WAM_Av._ Photolysis + Hydrolysis (RSD) [%]	PS Plant Sorption [%]
0	0.0 (7.3)	0.0 (1.6)	0.0
1	5.2 (0.9)	0.7 (2.5)	4.5
2	52.9 (7.6)	33.5 (1.4)	19.4
3	68.6 (5.4)	40.9 (1.6)	27.7
4	74.9 (7.1)	50.7 (7.4)	24.2
5	81.8 (3.9)	53.0 (1.7)	28.8
6	85.5 (8.7)	61.8 (1.6)	23.7
7	86.9 (6.7)	65.7 (5.4)	21.2
9	88.4 (2.5)	74.4 (2.0)	14.0
11	92.2 (6.7)	80.8 (4.1)	11.4
14	96.0 (8.0)	83.8 (1.6)	12.2

PS = PWAM_Av_ − WAM_Av._

**Table 2 ijerph-19-16994-t002:** Removal of TRI from water through photolysis, hydrolysis and sorption by the plant.

Day	Factors		
	PWAM_Av._ Photolysis + Hydrolysis + Plant Sorption (RSD) [%]	WAM_Av._ Photolysis + Hydrolysis (RSD) [%]	PS Plant Sorption [%]
0	0.0 (6.9)	0.0 (1.2)	0.0
1	9.0 (5.0)	1.6 (2.7)	7.4
2	41.6 (3.6)	5.1 (2.4)	36.5
3	63.1 (6.2)	7.5 (0.3)	55.6
4	66.2 (8.8)	13.5 (6.2)	52.7
5	68.5 (5.0)	14.2 (4.9)	54.3
6	70.3 (9.7)	18.8 (1.2)	51.5
7	70.8 (6.2)	19.4 (4.2)	51.4
9	72.2 (2.7)	24.1 (5.9)	48.1
11	73.5 (8.5)	33.6 (6.5)	39.9
14	75.4 (7.0)	37.4 (7.4)	38.0

PS = PWAM_Av_ − WAM_Av._

**Table 3 ijerph-19-16994-t003:** SMX and TRI content in plant tissue, with bioaccumulation factor (BAF).

Compound	Day	Average Concentration (SD) [ng g^−1^]_FW_	Average Concentration (SD) [ng g^−1^]_DW_	BAF_DW_	BAF_FW_
SMX	7	4.4 (0.1)	146.6 (3.2)	0.147	0.004
	14	1.3 (0.1)	42.6 (0.5)	0.043	0.001
TRI	7	78.4 (2.3)	2587.7 (113.3)	2.588	0.078
	14	72.0 (3.9)	2368.9 (148.1)	2.369	0.072

DW—dry weight; FW—fresh weight; BAF—bioaccumulation factor; SD—standard deviation.

**Table 4 ijerph-19-16994-t004:** TPs of TRI identified in plant and water during the 14-day experiment.

Abbrev.	[M + H]^+^ (*m*/*z*)	Water																						Plant	
		0 Day		1 Day		2 Day		3 Day		4 Day		5 Day		6 Day		7 Day		9 Day		11 Day		14 Day		7 Day	14 Day
		WAMs	PWAMs	WAMs	PWAMs	WAMs	PWAMs	WAMs	PWAMs	WAMs	PWAMs	WAMs	PWAMs	WAMs	PWAMs	WAMs	PWAMs	WAMs	PWAMs	WAMs	PWAMs	WAMs	PWAMs		
TRI171	171.0	-	-	-	-	-	+	-	+	-	+	-	+	-	+	-	+	-	+	-	-	-	-	-	-
TRI325	325.0	-	-	-	-	-	-	-	-	-	-	-	-	-	-	+	-	+	-	+	-	-	-	-	-
TRI291	291.1	-	-	-	-	-	-	-	-	-	-	-	-	-	-	-	-	+	-	+	-	+	-	-	-
TRI323	323.1	-	-	-	-	-	-	-	-	-	-	-	-	-	-	-	-	-	-	-	-	-	-	+	+
TRI305	305.2	-	-	-	-	-	-	-	-	-	-	-	-	-	-	-	-	-	-	-	-	-	-	-	+
TRI307	307.1	-	-	-	-	-	-	-	-	-	-	-	-	-	-	-	-	-	-	-	-	-	-	+	+

“+”—detected; “-”—not detected.

**Table 5 ijerph-19-16994-t005:** Transformation products of SMX identified in plant and water during the 14-day experiment.

Abbrev.	[M + H]^+^ (*m*/*z*)	Water																						Plant	
		0 Day		1 Day		2 Day		3 Day		4 Day		5 Day		6 Day		7 Day		9 Day		11 Day		14 Day		7 Day	14 Day
		WAMs	PWAMs	WAMs	PWAMs	WAMs	PWAMs	WAMs	PWAMs	WAMs	PWAMs	WAMs	PWAMs	WAMs	PWAMs	WAMs	PWAMs	WAMs	PWAMs	WAMs	PWAMs	WAMs	PWAMs		
SMX173	173.0	-	-	-	-	-	-	-	-	+	+	+	+	+	+	+	+	+	+	+	-	+	-	-	-
SMX256	256.1	-	-	-	-	-	-	-	-	-	-	+	-	+	-	+	-	+	-	+	-	+	-	-	-
SMX288	288.0	-	-	-	-	-	-	-	-	-	-	-	-	-	-	-	+	-	+	-	+	-	+	-	-
SMX304	304.1	-	-	-	-	-	-	-	-	-	-	-	-	-	-	-	-	-	-	-	-	-	-	+	+
SMX254	254.0	-	-	-	-	-	-	-	-	-	-	-	-	-	-	-	-	-	-	-	-	-	-	+	-
SMX445	445.1	-	-	-	-	-	-	-	-	-	-	-	-	-	-	-	-	-	-	-	-	-	-	+	+
SMX216	216.0	-	-	-	-	-	-	-	-	-	+	-	+	-	+	-	+	-	+	-	-	-	-	+	-
SMX179	179.4	-	-	-	-	-	-	-	-	-	-	+	-	+	+	+	+	+	+	-	+	-	+	-	+
SMX246	246.0	-	-	-	-	-	-	-	-	-	-	-	-	-	-	-	-	-	-	-	-	-	-	-	+

“+”—detected; “-”—not detected.

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
