# Peer review of "Study of the Phytoextraction and Phytodegradation of Sulfamethoxazole and Trimethoprim from Water by Limnobium laevigatum"

_ijerph, 2022, doi:10.3390/ijerph192416994_

Round 1

Reviewer 1 Report

The work is as correct and interesting. The idea of the research, its derivation, interpretation are also correct. However, I miss a clear indication of what is completely new in this work, what was done for the first time and why it is worth publishing. I would encourage the authors to supplement the publication test with this information. I disagree with the authors' statement presented in lines 457-458, as this is not the first publication in this area.

Extraction of both compounds from plant material is very simple. In my opinion, however, the sample is far from sufficiently purified. I think this may affect the quality of the results obtained and, therefore, also their reliability. Why didn't the authors additionally purify it with e.g. SPE, as they did with the water samples? If the authors disagree with my point of view, please provide evidence (results, chromatograms) to support their view.  

What was the dead time of the system?

I highly rate chapter 3.2. I think it is the best part of this work, in which the authors not only presented their own results but also related them to data presented so far in the literature. However, I understand that releasing TPs into water is environmentally friendly according to the authors?

Figure 2 and Figure 4: why the authors, out of all the signals on the spectra, chose for interpretation (and identification) only selected, and not the highest signals (but comparable in intensity to many others on the spectrum, which in turn were not taken into account)?

Author Response

Thank you very much for your critical review. It was very useful in the correction of our manuscript. Identification of weak points throughout the text has helped us to increase the value of our paper. All comments and changes suggested by Reviewer 1 have been incorporated into the manuscript. Once again,thank you very muchfor your help.

The responses are attached.

Reviewer 2 Report

L121 - variant 3 (PW) not mentioned in the text. What is its purpose?

L119 - Would each experiment be a repetition (PWAMs)? Why is there no repetition for the WAM variant (photolysis + hydrolysis)?

How is the plant sorption calculated? (PS = PWAM - WAM)? If yes, review Table 1, Day 3 => 68.6 - 40.9 = 27.7; Day 7 => 86.9 - 56.7 = 30.2. And review Table 2 in days 3, 4, 5, 9, and 11. Revise sorption values and days in discussion topic. Or explain how it was calculated.

Tables 1 and 2 for PWAM should present the mean removal of the three experiments +- standard deviation. For WAM treatment it would also be essential to have three repetitions and present mean +- standard deviation, to know the existing variation.

Table 2 - Day "55" => Day "5"

L222 - absorption or sorption??

Author Response

Thank you very much for your review. We are absolutely convinced that your additional comments significantly improved the scientific value of our paper. All comments and changes suggested by Reviewer 2 have been incorporated into the manuscript. Once again, thank you very much for your help.

The responses are attached.
